# Determining the Antenna Phase Center for the High-Precision Positioning of Smartphones

**DOI:** 10.3390/s24072243

**Published:** 2024-03-31

**Authors:** Fei Shen, Qianlei Hu, Chengkai Gong

**Affiliations:** 1School of Geography, Nanjing Normal University, Nanjing 210023, China; 221312029@stu.njnu.edu.cn; 2Key Laboratory of Virtual Geographic Environment, Nanjing Normal University, Ministry of Education, Nanjing 210023, China; 3Jiangsu Center for Collaborative Innovation in Geographical Information Resource Development and Application, Nanjing 210023, China; 4Jiangsu Basic Surveying and Mapping Facilities Technical Support Center, 28 Hongjun Street, Jiangxinzhou Street, Jianye District, Nanjing 210019, China; 191335003@stu.njnu.edu.cn

**Keywords:** GNSS antenna phase center, Xiaomi Mi8, relative positioning, high-precision positioning

## Abstract

In recent years, smartphones have emerged as the primary terminal for navigation and location services among mass users, owing to their universality, portability, and affordability. However, the highly integrated antenna design within smartphones inevitably introduces interference from internal signal sources, leading to a misalignment between the antenna phase center (APC) and the antenna geometric center. Accurately determining a smartphone’s APC can mitigate system errors and enhance positioning accuracy, thereby meeting the increasing demand for precise and reliable user positioning. This paper delves into a detailed analysis of the generation of Global Navigation Satellite System (GNSS) receiver antenna phase center errors and proposes a method for correcting the receiver antenna phase center. Subsequently, a smartphone positioning experiment was conducted by placing the smartphone on an observation column with known coordinates. The collected observations were processed in static relative positioning mode, referencing observations from geodetic-grade equipment, and the accuracy of the static relative positioning fixed solution was evaluated. Following weighted estimation, we determined the antenna phase center of the Xiaomi Mi8 and corrected the APC. A comparison of the positioning results of the Xiaomi Mi8 before and after APC correction revealed minimal impact on the standard deviations (STDs) but significant influence on the root mean square errors (RMSEs). Specifically, the RMSEs in the E/N/U direction were reduced by 59.6%, 58.5%, and 42.0%, respectively, after APC correction compared to before correction. Furthermore, the integer ambiguity fixing rate slightly improved after the APC correction. In conclusion, the determination of a smartphone’s APC can effectively reduce system errors in the plane direction of GNSS positioning, thereby enhancing smartphone positioning accuracy. This research holds significant value for advancing high-precision positioning studies related to smartphones.

## 1. Introduction

As of 2016, the original observations of the Global Navigation Satellite System (GNSS) can be extracted from Android Nougat (7.0) or later operating systems, which allows developers to optimize the navigation and positioning performance of smart terminals at a deeper level [1,2]. Since then, researchers have dedicated significant efforts to exploring the potential of Android smartphones, leading to the development of novel algorithms aimed at enhancing GNSS positioning performance for these widely used mass market devices.

In 2018, Xiaomi Mi8, the world ‘s first dual-frequency GNSS smartphone equipped with a Broadcom BCM47755 chipset, was launched [3,4]. This innovative device can track L1/E1 and L5/E5 codes and carrier phase signals from GPS, Galileo, and QZSS, as well as single-frequency measurements from GLONASS L1 code and BDS B1 code. Consequently, this groundbreaking smartphone has been extensively studied by scholars worldwide in the field of smartphone positioning. For instance, Sui Mingming et al. [5] introduced an optimized random model tailored to the characteristics of smartphone observations. Through this approach, they successfully enhanced the positioning accuracy of smartphones, contributing to the growing body of research on leveraging the capabilities of dual-frequency GNSS smartphones like the Xiaomi Mi8 smartphone. However, certain factors such as phase center offsets (PCOs) and the integer ambiguity resolution of smartphone phase observations were not considered. These elements are indeed crucial in refining positioning accuracy, especially in GNSS applications. Additionally, Warrant et al. [6] utilized the Android 7 or higher versions of smartphones to assess the potential of the Xiaomi Mi8 smartphone in achieving high-precision positioning through short baseline DGPS with a carrier smoothing code (Kalman filter). However, it is noteworthy that many earlier studies expressed skepticism regarding the feasibility of fixing the ambiguity of the carrier phase in smartphone GNSS observations. Consequently, experimental results often only considered the float solution, which typically yields sub-meter-level accuracy [7,8,9]. Paziewski et al. [10] evaluated the observational quality of the four navigation systems received by the mainstream Android smartphones’ GNSS antenna on the market comprehensively and systematically and analyzed the observation residuals by linear combination. Their experimental results showed that the pseudo-range observational noise was much larger than the carrier observational value, and the fixed solutions of the smartphone were obtained under static conditions, which verified the feasibility of centimeter-level positioning.

While the latest generation of GNSS chips boasts improved efficiency compared to their predecessors, the positioning accuracy of smartphones remains constrained by their inherent performance limitations. Consequently, numerous scholars have shifted their focus towards researching GNSS antennas tailored specifically to smartphones. In 2019, Netthonglang et al. [11] estimated the approximate APC of the Xiaomi Mi8 smartphone through average north and east coordinates. Their research showed that the relative positioning of smartphones could achieve centimeter-level accuracy after weakening the multipath effect. However, there was no effective method used to eliminate the multipath effect in the study and the amount of data collected was limited. In 2020, Bochkati et al. [12] used a choke ring antenna to suppress multipath effects from the ground and finally fixed the ambiguity correctly, obtaining a positioning accuracy of 2 cm. At the same time, the GNSS APC in the smartphone was also estimated. However, due to a lack of effective control of cycle slips and other factors in data processing, there were still some results that were intelligently float solutions, and the stability of positioning was poor. Wanninger et al. [13] successfully fixed the ambiguity and calibrated the phase center of the HUAWEI P30 smartphone. However, they believed that successful ambiguity fixing could only be performed on GPS L1 observations, that is, dual-frequency carrier phase ambiguity fixing was not feasible, which led to its multi-system dual-frequency observations not being fully utilized. In addition, they only performed relative antenna calibration for GPS L1, and the results showed that the horizontal phase center deviated from the midline of the smartphone. In 2021, Darugna et al. [14] used Geo++’s robot to perform absolute antenna calibration on a dual-band HUAWEI MATE20X smartphone and estimated the PCO and PCV. In addition, Google announced that starting from Android 11, Android phones can use the GnssAntennaInfo class to access GNSS antenna information. However, the correction information provided is only for a specific device model and is not perfect, and the reliability and compatibility of the GNSS antenna information still need to be improved and optimized.

In summary, current research efforts by scholars in the field of smartphone GNSS positioning primarily revolve around assessing smartphone GNSS observational quality and exploring GNSS positioning applications. However, there remain several areas warranting further investigation. The existing method for the GNSS antenna correction of smartphones relies on specialized and costly professional equipment, posing high entry barriers and consuming significant time, rendering it unsuitable for meeting the mass market demand for high-precision smartphone positioning. Moreover, neither domestic nor international software and hardware manufacturers have provided comprehensive information on GNSS antennas tailored for smartphones. The determination of APC in a smartphone can play a role in its manufacturing process. Through accurate design, manufacturing, and data processing, relatively accurate antenna phase center correction is realized on a mobile phone, thereby improving the positioning accuracy and reducing the error. Consequently, there is an urgent need for the development of a simple and efficient method for studying GNSS antenna correction information specifically designed for smartphones.

Aiming at the problem that the position of the GNSS antenna in smartphones is unknown at present, this paper first introduces the GNSS receiver antenna and its phase center. Then, the causes and characteristics of the phase center errors were analyzed. Finally, based on this, the positioning ability of the dual-frequency smartphone Xiaomi Mi8 was studied. The APC position of the Xiaomi Mi8 was estimated by a large number of high-precision positioning results of the smartphone. Subsequently, the phase center deviation of the smartphone antenna was calibrated and rectified, with the antenna correction information integrated into the solution strategy for recalculating the positioning results. Through a comparative analysis of the smartphone’s positioning accuracy before and after the APC correction, this study confirms the efficacy of APC correction in enhancing a smartphone’s positioning accuracy.

## 2. Methods

This section offers an overview of the fundamental theory underlying a GNSS receiver’s antenna phase center (APC) and presents a detailed method for determining the APC of a smartphone.

### The Theory of APC Correction

The correction of GNSS receiver phase center errors was proposed as early as the mid-1980s (Sims, 1985). GNSS antenna phase center correction includes mean phase center offsetting with respect to the antenna reference point (ARP) or the mass of a satellite, i.e., phase center offsets (PCOs), and the variation in the mean phase center with respect to the satellite elevation and azimuth angle, i.e., phase center variations (PCVs) [15]. The APC is a virtual point, which refers to the actual position where the antenna receives a GNSS satellite signal. The ideal receiving antenna has a unique and fixed phase center. However, when the antenna receives the electromagnetic wave signals from satellites in different directions, the additional phase difference caused by the antenna itself will cause a deviation in the distance measurement results, thus introducing the errors of distance measurement results. The PCV of an antenna refers to the gap between the instantaneous APC and the average APC. The ARP refers to the intersection of the vertical symmetry axis of an antenna and the bottom surface of the antenna. The instantaneous APC changes with the elevation angle and azimuth angle of the received signal, so the PCV can be described as a function of the elevation angle and azimuth angle ${∆}_{PCV}\left(a,z\right)$:(1)$${∆}_{PCV}\left(a,z\right)=\sum_{n=0}^{n_{max}}\sum_{m=o}^{n}\left[A_{nm}\cos\left(ma\right)+B_{nm}\sin\left(ma\right)\right]P_{nm}\cos\left(z\right)$$
where a and z refer to the position of a specific satellite in the antenna coordinate system, $P_{nm}$ is a fully normalized Legendre polynomial, and $A_{nm}$ and $B_{nm}$ are the estimation coefficients of the maximum order $n_{max}$ and the maximum order $m_{max}<n_{max}$, respectively.

Figure 1 shows the schematic diagram of the receiver antenna phase center. The correction term of the phase center of the receiver antenna in the ECEF coordinate system is $d_{r}$:(2)dr=ErT(dr,pco+dr,APC)+dr,pcv
where $E_{r}$ is the conversion matrix between the ECEF coordinate system and the station center coordinate system and $d_{r,pco}$ and $d_{r,pcv}$ are the antenna PCO and variations’ PCV in the station center coordinate system, respectively. At present, the calibration of a GNSS antenna’s PCO/PCV is mainly monopolized by foreign institutions, and the cost is high. In addition, due to the inferior quality of smartphone observations, it is extremely sensitive to multipath effects, and the calibration of the antenna needs to be modeled separately to eliminate the influence of multipath effects. Using an anechoic chamber [16] or an automatic rotating robot [17] to calibrate a smartphone GNSS antenna can become extremely complicated and time-consuming. Therefore, only the APC was studied here, that is, the correction term $d_{r,APC}$ in Equation (2).

For the GNSS antenna of a geodesic receiver, the phase center varies by less than a few millimeters with the satellite azimuth or elevation angle of the received signal. However, the value of the smartphone GNSS will be much larger, because most smartphone designers are not experts in GNSS and RF antennas. Smartphone designers usually choose cheap GNSS antennas and embed them in the smartphone shell for cost-effectiveness and compression volume considerations when designing GNSS antennas. The location is most susceptible to interference from other signal sources [18]. Generally, GNSS engineers believe that GNSS antennas should be as far away from Bluetooth, Wi-Fi, radio, and other signal transmission equipment as possible. However, due to the highly integrated antenna design inside a smartphone, the interference of the internal signal source cannot be avoided, resulting in the APC usually being inconsistent with the antenna’s geometric center. This point may even be outside the phone shell. For the purpose of high-precision positioning, the APC must be determined. In this section, a large number of experiments were conducted to determine the position of the APC relative to the smartphone geometry.

Based on the problem of drastic changes in the APC of a smartphone, a simple method is proposed to determine the position of the APC. The premise of this method is to require the smartphone observation to be able to solve the absolute coordinates of the centimeter-level accuracy. Secondly, it is stipulated that the millimeter-level accuracy reference point coordinates used and the calculated centimeter-level accuracy smartphone coordinates are in the same geodetic reference system. In addition, multipath errors are another key factor affecting the positioning quality in mobile GNSS. The cheap antenna in the smartphone does not have the ability to suppress multipath errors. Therefore, the choke platform was selected to resist the multipath errors of the smartphone, and a measurement area with open sky conditions was selected to ensure that enough satellites could be tracked, especially satellites with L5/E5a signals (GPS and Galileo), because they play a key role in fixed ambiguity [10,19]. Since the real position of the smartphone had to be known very accurately, an observation column with exactly accurate coordinate information was used as a reference. Figure 2 shows a schematic of the Xiaomi Mi 8 smartphone and the choke coil used to suppress multipath errors. The red dot position in the figure is the smartphone ARP. The direction of the smartphone charging port to the midpoint of the top of the fringe was used as the positive direction of the smartphone.

Furthermore, the deviation resulting from the volume of the choke in the real coordinates of the observation column was taken into account during the positioning correction process. To validate the effectiveness and applicability of this experiment, a total of 12 sets of data were collected on 18 October, 15 November, and 6 December 2021. Following the research conducted by Bochkati et al. [12], the optimal satellite distribution state typically persists for approximately 1–1.5 h. Therefore, each set of data was recorded for more than 1 h to guarantee the robustness and reliability in the subsequent analysis. Considering the low-cost linear polarization antenna of the smartphone, it is important to note that the satellite states tracked may vary slightly across different azimuth angles. To mitigate this variability, prior to each data acquisition session, the smartphone orientation was adjusted with the red dot in Figure 2 serving as the rotation center. This adjustment was made to ensure that the experimental outcomes remained unaffected by satellite azimuth changes. Finally, the experimental results of a total of more than 18 h were statistically analyzed. A flow diagram of the experimental methodology of this paper is depicted in Figure 3.

## 3. Collection and Analysis of Data

This section aims to introduce the data collection environment utilized in this experiment and provide a brief analysis of the collected data.

### 3.1. Data Collection

In this study, GNSS observations were conducted using two Xiaomi Mi8 smartphones (one black and one white) and a HUAWEI MATE20 smartphone. The data collection took place on the roof of the School of Geography, Nanjing Normal University, with a sampling interval of 1 s. A total of 4.5 h of observation data were acquired between 1:30 and 6:00 UTC on 15 November 2021. To facilitate the observations, three smartphones were positioned on a sky barrier-free observation column. Additionally, a Trimble Alloy geodetic receiver was strategically placed approximately twenty meters away from the smartphones. This receiver was connected to the Trimble GNSS-Ti choke ring antenna, serving as the reference station for this experiment. Figure 4 illustrates various specific experimental equipment and their surrounding conditions. The reference coordinates of the observational column in the figure were precisely determined by employing the geodetic receiver to record long-term static observations.

Firstly, a Xiaomi 8 mobile phone was positioned on the choke antenna platform of the No. 4 observation column, while another Xiaomi 8 and a Huawei Mate 20 mobile phone were positioned on the adjacent No. 3 and No. 5 observation columns, respectively. To enhance clarity in expression, “XM8B” and “XM8W” will denote two distinct colors of the same model of the Xiaomi smartphone, while “HW20” will refer to the Huawei Mate 20 smartphone in the following text. The choke ring antenna model used was the LEICA AR25, which is widely used by IGS, CORS, and other high-level observation stations because of its excellent anti-multipath effect. Since relative positioning requires a reference station with known coordinates, Trimble Alloy was used as a reference station to install on another observation column in all time periods. The baseline of the rover station and the reference station was about 20 m, and the real coordinates of the two reference stations were determined by the observation column. Because of this ultra-short baseline, the atmospheric delay was the same, so the delay was offset in the relative positioning, which helped to improve the ambiguity fixing rate and shortened the convergence time. Raw measurement data from both the reference station and from the smartphone were evaluated in post-processing using RTKLIB. We carried out secondary development of RTKLIB and optimized it for low-cost receivers and practical applications. More robust strategies were added to the program to deal with the problems caused by low-quality observations. In terms of the ambiguity processing strategy, the LAMBDA method was used to fix the ambiguity to an integer value.

### 3.2. Data Analysis

In addition, GPS results were computed for further comparison. Regarding the selection of Android GNSS observation data recording software, we prioritized stability and compatibility. After extensive testing, Geo++ RINEX logger emerged as the top performer and boasted the highest number of downloads in the application market. Consequently, we opted for Geo++ RINEX logger to record observation data. It should be noted that the HW20 smartphone in our experiment was incapable of outputting carrier phase observations due to some bug in the Harmony OS. Continuous carrier phase observations play a pivotal role in high-precision positioning technology. Although both smartphones were not affected by the duty cycle mechanism, Xiaomi Mi8 smartphones proved to be the optimal choice for high-precision positioning in this experiment.

To analyze the tracking ability of different devices to GNSS satellites, we distinguished different satellite systems and frequencies. Figure 5 shows the number of satellites observed at the L1/E1/B1/G1 frequencies, while Figure 6 shows the corresponding results at the L5/E5a frequencies. Table 1 lists the equipment used in the experiment and the GNSS signal tracking performance. Establishing a singular phase center for the L1, L5, E1, E5a, B1, and G1 bands across all theta and phi angles presented considerable challenges. Signal propagation and antenna characteristics vary significantly across different frequency bands and angles. Therefore, in our experiment, we propose to calculate an average phase center deviation specifically for the L5/E5a bands. This approach is particularly crucial as these bands play a pivotal role in ambiguity resolution. It should be noted that the Huawei smartphones were unable to output carrier phase observations because their operating system was replaced by Harmony OS. We could find two Xiaomi Mi8 smartphones equipped with the same GNSS chip, one of which could not however observe the BDS and Galileo satellites during the experiment. Since the Geo++ RINEX logger versions of all smartphones were the same, the number of tracking signals could have initially reflected the quality of the observations, which shows that the signal quality of the same smartphone was different in similar observation environments. In addition, although HW20 observed the largest number of satellites, considering that it lacked carrier phase observations, continuous carrier phase observations played a decisive role in high-precision positioning technology. Although both smartphones are not affected by the duty cycle mechanism, for this experiment, it was considered that the XM8B smartphone had the most potential for high-precision positioning, and XM8B became the best choice for achieving the high-precision positioning of smartphones. We will focus on XM8B for the next experimental analysis.

Table 1 lists the smartphones used in the experiment and their GNSS signal tracking performance, where √ indicates that the corresponding satellite signal can be observed, – indicates that the corresponding satellite signal cannot be observed.

## 4. Results

In this section, we determined the APC of the XM8B through the static relative positioning results derived from multiple sets of experimental data. Subsequently, we recalculated the positioning accuracy after applying APC correction and compared it with the accuracy observed prior to the recalculation.

### 4.1. Determination of Smartphone’s APC

Figure 7 shows the positioning errors of the static relative positioning calculated by the two sets of data collected on 18 October 2021, in the station center coordinate system. In the figure, STD reflects the standard deviations of the positioning result, and RMSE represents the root mean square errors, reflecting the difference from the real coordinates. However, due to the different APC positions of the two sets of data, the RMS value cannot truly reflect the positioning accuracy. In this particular case, subsequent analysis will use the STD to compare accuracy. From the results of the map, it can be seen that a higher ambiguity fixing rate brought high precision, and the positioning of the smartphone could achieve centimeter-level or even millimeter-level accuracy, because after the smartphone correctly fixed the ambiguity, the coordinate accuracy of the solution was determined by the accuracy of the phase observation.

Table 2 presents the statistical outcomes of the fixed solution station center coordinate system for all positioning scenarios. In the table, STD denotes the standard deviation of the fixed solution, and RMS represents the root mean square errors of the fixed solution. The “Orientation” column specifies the smartphone’s orientation, with terms like “WEST” indicating that the smartphone was oriented to the geomagnetic west. Observing the success rate of ambiguity fixing in the table, it was found that there was no robust correlation between the fixing rate and the positioning accuracy. For instance, the data from the 6 December 2021 SOUTH group exhibited the lowest STDs of 0.001, 0.001, and 0.003 m in the E/N/U directions, with a fixing rate of 100%. In contrast, the 15 November 2021 EAST group, with the lowest fixation rate of 54%, demonstrated STD values of 0.005, 0.004, and 0.014 m in the three directions, respectively. Surprisingly, these values were even smaller than the corresponding STD values of 0.006, 0.005, and 0.013 m in the SOUTHWEST group and 0.005, 0.009, and 0.0019 m in the NORTHWEST group on the same day, despite their fixing rates being 98.9% and 100%, respectively. Thus, the higher-ambiguity fixing rate not only enhanced the reliability of the experimental results but also validated that the GNSS observations collected by the XM8B, when placed on a choke ring, exhibited high-precision positioning capabilities, thereby fulfilling the requirements for antenna phase center determination.

To visually represent the relative positioning between the results and the XM8B device, Figure 8 illustrates the smartphone coordinate system. The diagram’s coordinate system is structured as the following: the ARP of the XM8B serves as the origin, the Y axis extends from the smartphone’s charging port to the midpoint of the fringe, the X axis is perpendicular to the Y axis, and the Z axis is perpendicular to the plane formed by the X and Y axes. The positive direction of the coordinate system adhered to the right-hand rule.

In Figure 9, we present two sets of point cloud top views within the smartphone coordinate system, calculated using data from 18 October 2021, with one set depicting the WEST orientation (left) and the other showing the SOUTH orientation (right). Within the figures, the red dots represent the fixed solution set derived from static relative positioning achieved by the smartphone. The green cross is indicative of the estimated APC of the XM8B, and the average value of the sample that obeyed the two-dimensional Gaussian distribution and was in the 95% confidence interval (2σ) was used as the APC. Figure 10 displays the disassembly diagram of the XM8B smartphone, with the backside (screen facing down) depicted. It should be noted that the schematics of all smartphones above have the screen up. The top metal patch of the red virtual coil in the figure is the GNSS antenna of the smartphone. Ideally, it is assumed that the APC of a smartphone’s GNSS should be presumed to coincide with the metal patch. To substantiate this hypothesis, numerous experiments were conducted in this section. During these experiments, when the position of the red dotted wireframe in Figure 8 was obstructed, there was a rapid decay in the signal strength of the GNSS. A comparative analysis between Figure 7 and Figure 8 revealed that the geometric center of the XM8B smartphone GNSS antenna did not align with the actual phase center.

To refine the determination of the antenna phase center (APC) position, Table 3 enumerates the phase center coordinates (X, Y, Z) in the smartphone coordinate system derived from all 12 sets of data. Almost all the coordinates fell in the eighth quadrant of the coordinate system, near the origin. Notably, the majority of coordinates resided within the eighth quadrant of the coordinate system, proximate to the origin. Furthermore, it is evident that the APC error minimally impacted the horizontal precision positioning, which remained at the centimeter level, while demonstrating a decimeter-level effect on the elevation direction. Combining the information from Table 2, the STD of each group of data served as the weight matrix, and the weighted average of the coordinates of the corresponding three directions in Table 3 was calculated. Finally, the APC coordinates in the smartphone coordinate system were –0.024, –0.028, and –0.116, with units in meters. Since then, the APC of the XM8B smartphone could serve to provide deviation correction information. It is imperative to accurately ascertain the position of an APC to lend reference significance to calculated high-precision coordinates.

Figure 11 illustrates the schematic diagram of the APC of the XM8B. It can be seen that there are a few centimeters of a gap between the actual phase center and the geometric center of the GNSS antenna of the XM8B smartphone. This discrepancy is attributed to the integration of additional signal receiving and transmitting functions within the GNSS antenna of the smartphone, rendering it susceptible to interference from other signal sources.

### 4.2. Verification of APC Correction

To further substantiate the effectiveness of APC correction, we incorporated the estimated antenna correction information into the solution strategy and subsequently recalculated the positioning results of the 12 sets of data, as shown in Table 4.

To analyze the results before and after the APC correction, Figure 12 compares the positioning accuracy depicted in Table 2 and Table 4 through broken lines. The following conclusions can be drawn from the analysis of the chart. Initially, considering the ambiguity fixing rate, it is noteworthy that the values for the remaining data groups exhibited remarkable proximity. However, it was observed that the fixing rates for Group 5 and Group 6 were 99.7% and 83.3%, respectively, surpassing the pre-correction values of 91.1% and 76.4%. Secondly, the statistical results of the STD in the left half of Figure 10 indicate that the APC correction had a negligible effect on the STD (internal coincidence accuracy), and the comparison illustrates that the two sets of broken lines almost coincided, indicating minimal disparity between the data points before and after the APC correction. The statistical results of the RMSE in the right half show that the APC correction had an enormous influence on the RMSE (external coincidence accuracy). The broken line (blue), corrected in the N/E direction, is closer to the 0 value than the (red) before the correction. After correction, the RMSE in the N/E direction experienced changes within the ranges of 0.003–0.037 m and 0.002–0.038 m, respectively. In contrast, the corresponding values before correction ranged from 0.009 to 0.055 m and 0.007 to 0.057 m, respectively. Consequently, the former range was approximately 0.006–0.037 m smaller than the latter in the plane direction. Remarkably, this difference closely aligns with the antenna phase center correction value.

It is noteworthy that we assessed the experiment’s significance solely through the relative positioning results before and after correction. Since there was no absolute truth serving as a reference, we cannot provide the absolute accuracy of the APC. Moreover, there were various errors, for example, stemming from the smartphone design, manufacture, and external environmental interference, which affected our APC accuracy. Furthermore, the APC correction process itself may have entailed measurement and correction errors. Nonetheless, we believe our work still holds some reference significance.

In short, the correction of a smartphone’s APC can effectively reduce the system error in the N / E direction in GNSS positioning, thereby enhancing the positioning accuracy.

## 5. Conclusions

In this paper, we conducted a detailed analysis of the causes behind the phase deviation of smartphone antennas in the current state of smartphone positioning. Multiple sets of positioning experiments were executed in an open static environment, employing three dual-frequency smartphones (XM8B, XM8W, and HW20). The Android application Geo++ RINEX logger facilitated observational recording. Initially, this paper evaluated the GNSS signal tracking capability of the experimental devices. Finally, XM8B, possessing the highest potential for high-precision positioning, was selected for subsequent experimental analysis.

In the experiment, Trimble Alloy served as the reference station, while the smartphone functioned as the rover station. With an ultra-short baseline of approximately 20 m, atmospheric delay was effectively eliminated during the experiment. Additionally, a choke ring was employed to mitigate multipath errors associated with the smartphone. Subsequently, the actual coordinates of the XM8B’s APC were determined based on the results obtained from multiple sets of static relative positioning. Furthermore, analyzing the variation amplitude of the APC in the smartphone coordinate system revealed that the line phase center exerted minimal influence on the horizontal direction of the precision positioning, achieving centimeter-level accuracy, while its impact on the elevation direction was at the decimeter level. Then, we corrected the APC and made a comparison between the experimental results obtained before and after the antenna correction. It was found that the APC correction had little effect on the STD (internal coincidence accuracy). The standard deviation difference in the N/E direction before and after the correction changed within a range of ±0.001 m, and the corresponding difference in the U direction changed within 0–0.004 m. Statistical analysis of the RMSE revealed a significant influence of the APC correction on the external coincidence accuracy, and the broken line after the N/E direction correction was closer to the zero value than that before the correction. Specifically, the RMSEs in the E/N/U direction decreased by 59.6%, 58.5%, and 42.0%, respectively, after the APC correction. Furthermore, the rate of integer ambiguity fixing slightly increased following the APC correction. In summary, through multiple sets of experiments, we confirmed the feasibility of determining the phase of a smartphone antenna to improve its positioning accuracy.

## Figures and Tables

**Figure 1 sensors-24-02243-f001:**
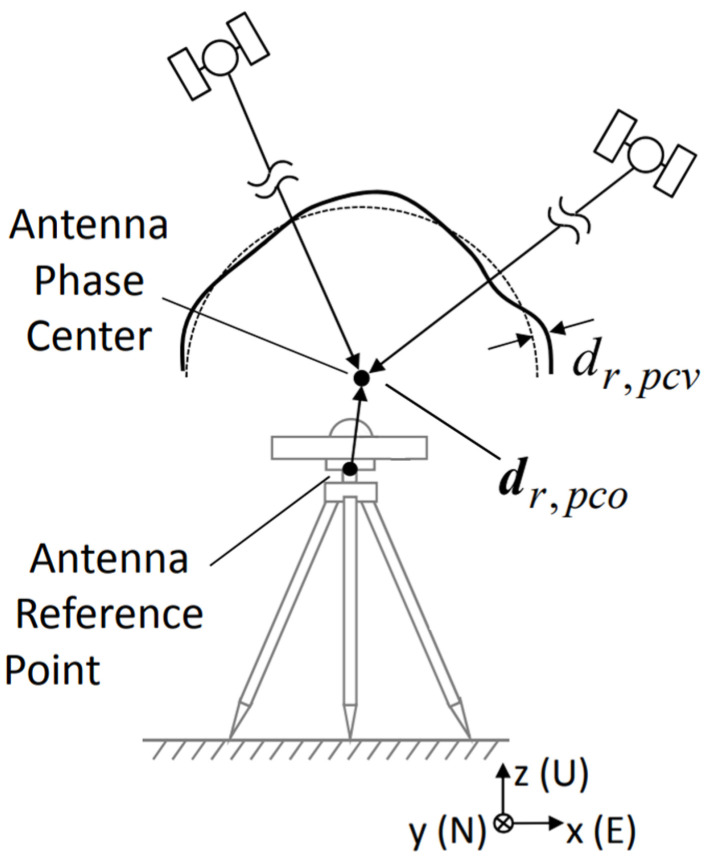
GNSS receiver antenna phase center.

**Figure 2 sensors-24-02243-f002:**
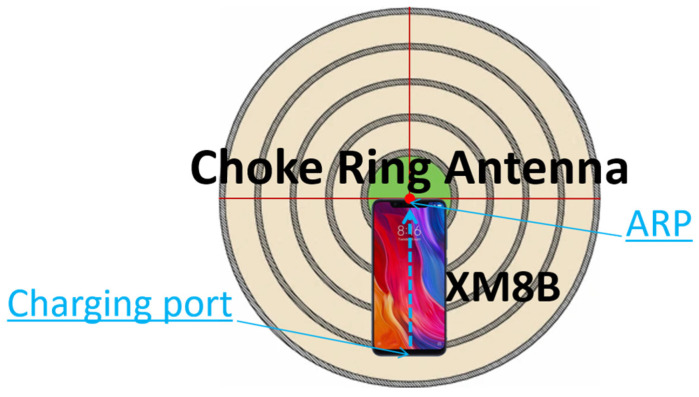
Experimental setup of smartphone antenna phase measurement.

**Figure 3 sensors-24-02243-f003:**
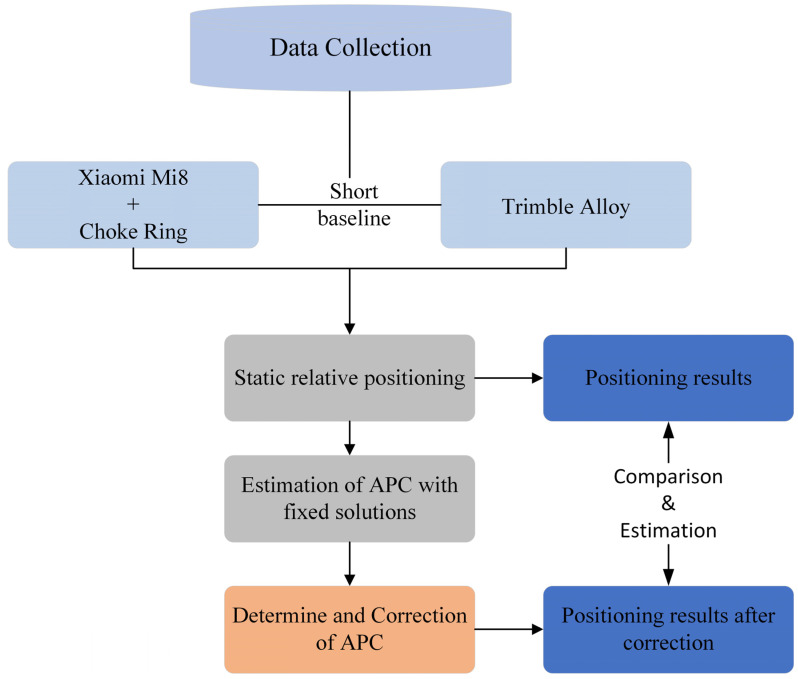
Flow diagram of the experimental methodology.

**Figure 4 sensors-24-02243-f004:**
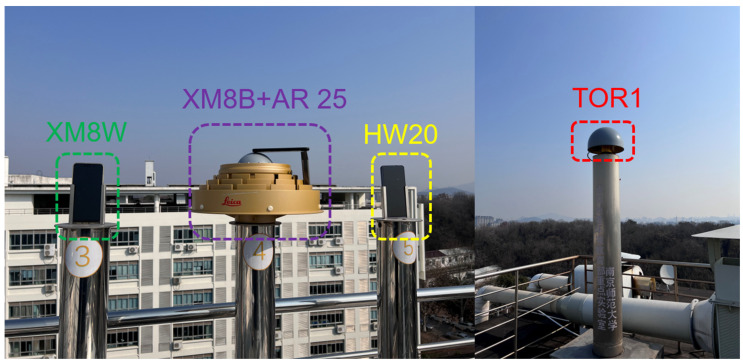
Surrounding environment and experimental equipment.

**Figure 5 sensors-24-02243-f005:**
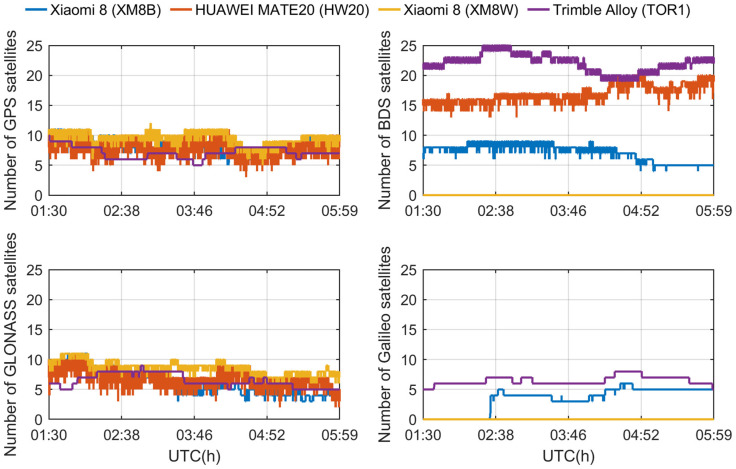
The number of satellites tracked by different receivers (corresponding to L1, E1, B1, and G1 band signals, respectively).

**Figure 6 sensors-24-02243-f006:**
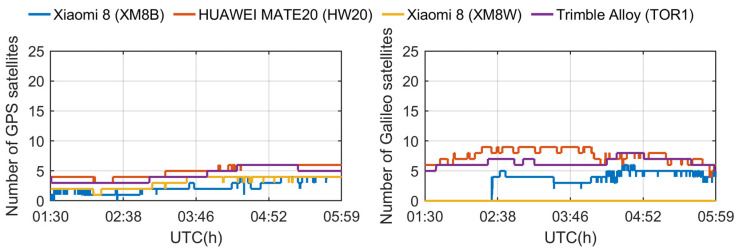
The number of satellites tracked by different receivers (corresponding to L5 and E5 band signals, respectively).

**Figure 7 sensors-24-02243-f007:**
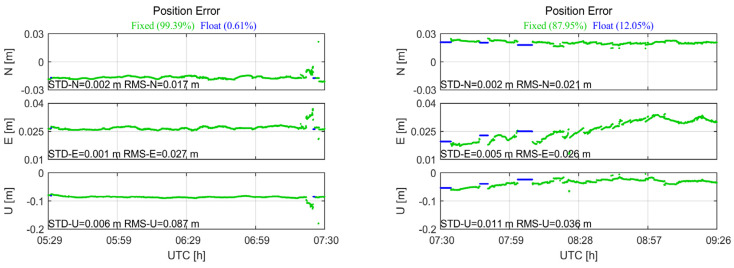
The positioning errors of the static relative positioning calculated by the two sets of data collected on 18 October 2021.

**Figure 8 sensors-24-02243-f008:**
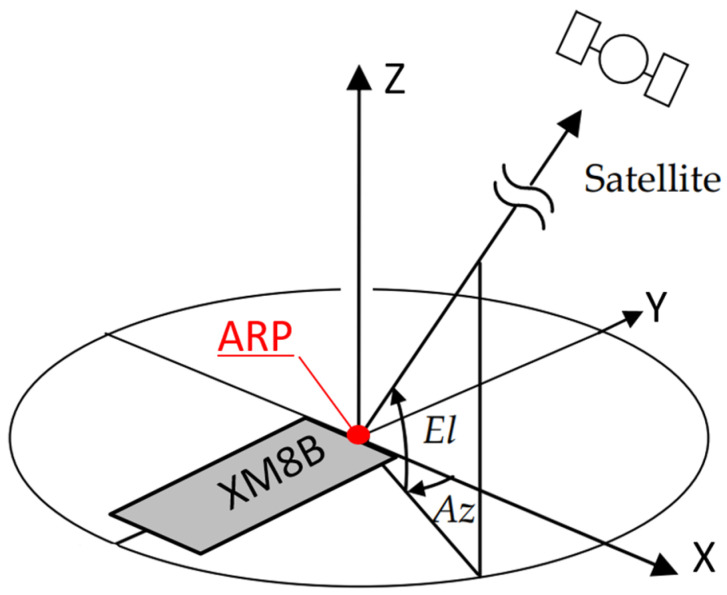
XM8B smartphone coordinate system.

**Figure 9 sensors-24-02243-f009:**
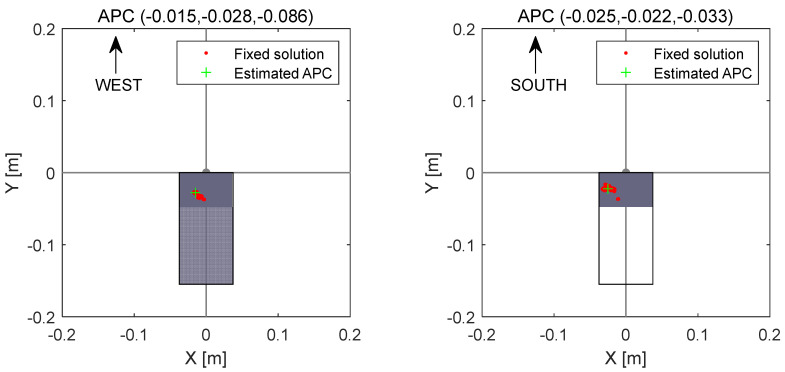
Point cloud top view in two sets of smartphone coordinate systems.

**Figure 10 sensors-24-02243-f010:**
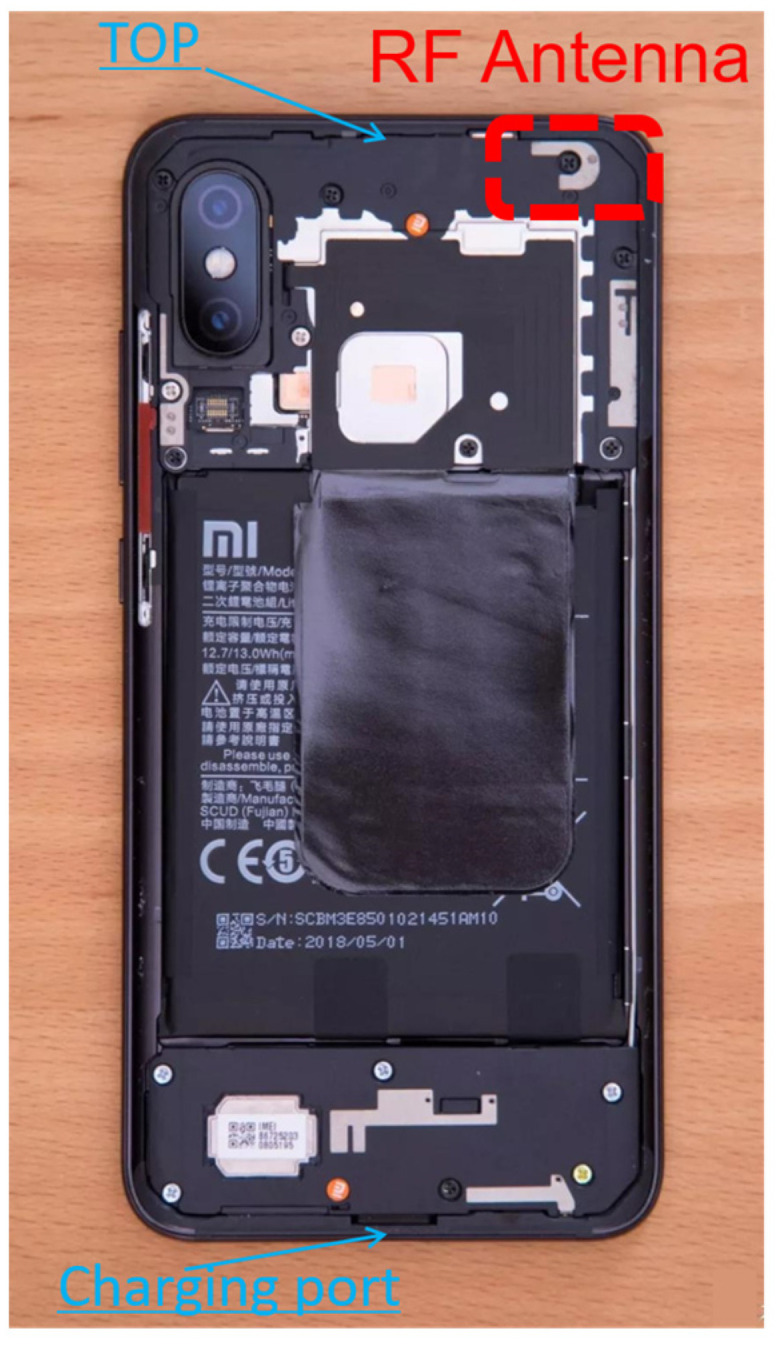
XM8B smartphone back (screen facing down) disassembly diagram.

**Figure 11 sensors-24-02243-f011:**
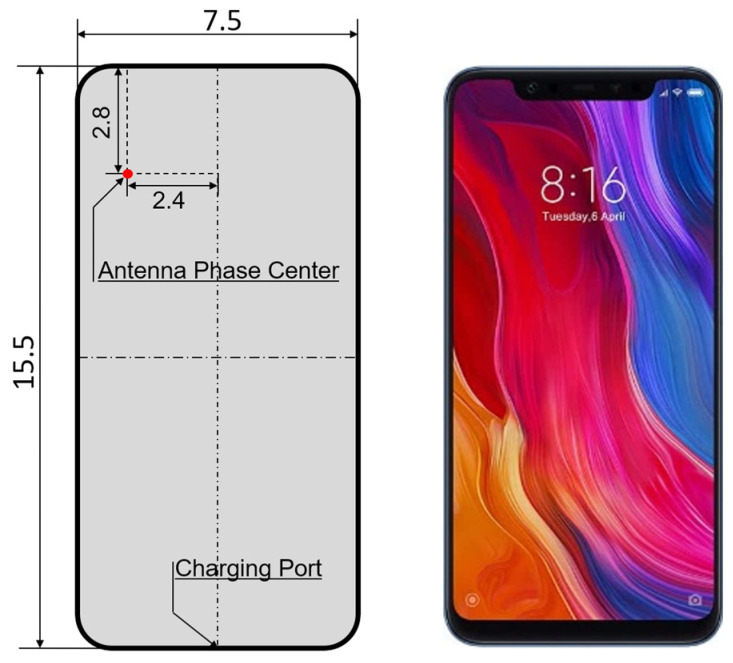
The estimated phase center position of the XM8B smartphone. (All dimensions are reported in cm).

**Figure 12 sensors-24-02243-f012:**
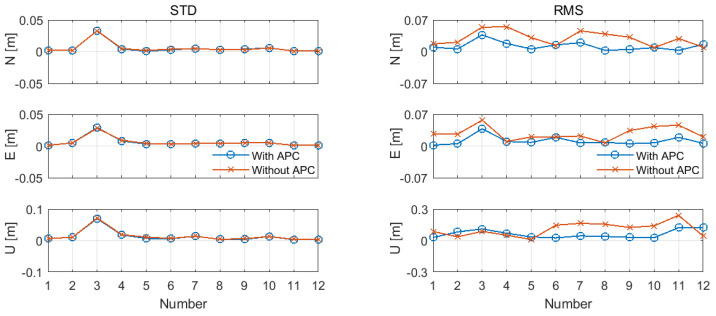
Precision statistical line charts before and after APC correction of the 12 sets of data.

**Table 1 sensors-24-02243-t001:** GNSS signal tracked by different smartphones.

Smartphone			GPS				Galileo				BDS		GLONASS	
Device	Chipset	RINEX ID	L1		L5		E1		E5a		B1		G1	
			L	P	L	P	L	P	L	P	L	P	L	P
Xiaomi 8	Broadcom BCM47755	XM8B	√	√	√	√	√	√	√	√	√	√	√	√
Xiaomi 8	Broadcom BCM47755	XM8W	√	√	√	√	–	–	–	–	–	–	√	√
HUAWEI MATE20	Kirin 980	HW20	–	√		√	–	–	–	√	–	√	–	√

**Table 2 sensors-24-02243-t002:** Statistics of static relative positioning results of 12 sets of data collected by the XM8B.

Date	Orientation	Number	Coordinate Statistics	North [m]	East [m]	Up [m]	Fixed Rate
18 October 2021	WEST	#1	STD	0.002	0.001	0.006	99.4%
			RMS	0.017	0.027	0.087	
	SOUTH	#2	STD	0.002	0.005	0.011	88.0%
			RMS	0.021	0.026	0.036	
15 November 2021	WEST	#3	STD	0.033	0.028	0.072	72.6%
			RMS	0.054	0.057	0.088	
	NORTHWEST	#4	STD	0.005	0.009	0.019	100.0%
			RMS	0.055	0.010	0.051	
	NORTHE	#5	STD	0.002	0.004	0.011	91.1%
			RMS	0.031	0.020	0.011	
	NORTHEAST	#6	STD	0.003	0.002	0.005	76.4%
			RMS	0.014	0.020	0.147	
	EAST	#7	STD	0.005	0.004	0.014	59.9%
			RMS	0.046	0.022	0.165	
	SOUTHEAST	#8	STD	0.003	0.004	0.004	99.8%
			RMS	0.039	0.007	0.156	
	SOUTH	#9	STD	0.003	0.005	0.006	100.0%
			RMS	0.032	0.034	0.125	
	SOUTHWEST	#10	STD	0.006	0.005	0.013	98.9%
			RMS	0.009	0.043	0.141	
16 December 2021	NORTH	#11	STD	0.001	0.001	0.004	99.4%
			RMS	0.029	0.046	0.241	
	EAST	#12	STD	0.001	0.001	0.003	100.0%
			RMS	0.010	0.020	0.042	

**Table 3 sensors-24-02243-t003:** The coordinates of the XM8B antenna phase center in the mobile coordinate system.

Period	Orientation	Number	X [m]	Y [m]	Z [m]
18 October 2021	WEST	#1	−0.015	−0.028	−0.086
	SOUTH	#2	−0.025	−0.022	−0.033
15 November 2021	WEST	#3	−0.025	−0.037	−0.015
	NORTHWEST	#4	−0.037	−0.039	−0.044
	NORTH	#5	−0.017	−0.032	−0.001
	NORTHEAST	#6	0.001	−0.025	−0.149
	EAST	#7	−0.044	−0.025	−0.166
	SOUTHEAST	#8	−0.021	−0.033	−0.156
	SOUTH	#9	−0.032	−0.033	−0.123
	SOUTHWEST	#10	−0.022	−0.036	−0.138
06 December 2021	NORTH	#11	−0.044	−0.032	−0.242
	EAST	#12	−0.009	−0.021	−0.042

**Table 4 sensors-24-02243-t004:** Statistics of the static relative positioning results of the 12 sets of data after APC correction.

Date	Orientation	Number	Coordinate Statistics	North [m]	East [m]	Up [m]	Fixed Rate
18 October 2021	WEST	#1	STD	0.002	0.001	0.006	99.4%
			RMS	0.010	0.002	0.030	
	SOUTH	#2	STD	0.002	0.005	0.011	88.0%
			RMS	0.006	0.005	0.083	
15 November 2021	WEST	#3	STD	0.033	0.029	0.070	72.7%
			RMS	0.037	0.038	0.111	
	NORTHWEST	#4	STD	0.004	0.008	0.018	100.0%
			RMS	0.018	0.010	0.070	
	NORTHE	#5	STD	0.001	0.003	0.007	99.7%
			RMS	0.006	0.008	0.035	
	NORTHEAST	#6	STD	0.003	0.003	0.006	83.3%
			RMS	0.015	0.019	0.026	
	EAST	#7	STD	0.005	0.004	0.014	59.9%
			RMS	0.020	0.007	0.045	
	SOUTHEAST	#8	STD	0.003	0.004	0.004	99.9%
			RMS	0.003	0.008	0.040	
	SOUTH	#9	STD	0.004	0.005	0.004	100.0%
			RMS	0.005	0.005	0.031	
	SOUTHWEST	#10	STD	0.006	0.005	0.013	98.9%
			RMS	0.009	0.007	0.027	
16 December 2021	NORTH	#11	STD	0.001	0.001	0.003	99.4%
			RMS	0.003	0.019	0.124	
	EAST	#12	STD	0.001	0.001	0.003	100.0%
			RMS	0.016	0.020	0.042	

## Data Availability

Data are contained within the article.

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
