# Peer review of "Determining the Antenna Phase Center for the High-Precision Positioning of Smartphones"

_sensors, 2024, doi:10.3390/s24072243_

Round 1
Reviewer 1 Report
Comments and Suggestions for Authors
Authors present an interesting topic of improving the positioning accuracy of a smartphone using the antenna phase center information. After careful review, this reviewer finds some series issues that need to be resolved.
1. Figure 4 shows the geometrical setup for APC measurements. It is well known the antenna characteristics including the phase center is affected by the structure around the antenna. In this regard, an ideal antenna for APC correction is a geodesy-grade choke ring antenna installed at a fixed position some height above the ground. For mobile phones, it might not be practical to use APC correction since the antenna surroundings change constantly. The idea of APC correction for the positioning with a smartphone needs to be reconsidered.
2. Antenna phase center is a function of the incoming wave direction and the frequency. For example, see R. Zhou et al., "Absolute field calibration of receiver antenna phase center models for GPS/BDS-3 signals", J Geodesy, 97, 2003.
https://link.springer.com/article/10.1007/s00190-023-01773-7
It seems that authors try to find a single phase center for L1, L5, E1, E5a, B1 and G1 (spanning 1164-1610MHz) and for all theta and phi angles.
The validity of using a single APC for all angles and all satellites needs to be rechecked.
3. The accuracy of the measured APC and the sources of errors on the measured APC should be given.
4. This reviewer questions the use of Table 4 where relative positioning data after APC correction. Absolute position accuracy (in GPS earth coordinates and in cm) before and after correction should be used.
Author Response
Thank you for your comments. I have uploaded an attachment. Please see the attachment.

Reviewer 2 Report
Comments and Suggestions for Authors
This paper explores the determination of phase centre deviation of smartphone GNSS antennas. The above research is innovative to a certain extent, but it still needs some modifications before publication. Here are some suggestions and questions:
1) In Figure 1, the location pointed to by "Antenna Phase center" should be wrong. It should be the instantaneous phase center.
2) The symbol Ac in line L149 is not seen in Figure 1.
3) What is dr,APC of formula (2) in line 150?
4) How to get "the phase center coordinates" in Table 3?
5) Many expressions are repeated in this paper. It is recommended to carefully check and streamline unnecessary repetitions.
6) The expression in line 138 should be incorrect and difficult to understand.
7) The deviation determined in this paper should be the average phase centre deviation of the antenna(PCO), not the antenna phase centre(APC), which changes instantaneously.
8) “a sampling rate of 1 second.” should be“a sampling interval of 1 second.”
9) "Huawei smartphones are incapable of outputting carrier phase observations due to Harmony OS replacing the operating system " is not rigorous. Because there are some equipped with Harmony OS are able to outputting carrier phase observations. Similarly, there are Android smartphones which are incapable of outputting carrier phase observations
10) "Botong" should be "Broadcom".
11) In line 319, should "Figure 6" be "Figure 8"?
12) In line 332, should "Figure 8" be "Figure 10"?
13) In line 380, should "Figure 10" be "Figure 12"?
14) In " 4.2. Verification of APC correction ", the data used are the same as those used to calculate the APC deviation, which makes the final result a bit unconvincing. If possible, it is recommended to add a set of examples from other scenarios for verification.
15) The conclusions are a bit wordy and they should be streamlined appropriately
Comments on the Quality of English Language1) Many expressions are repeated in this paper. It is recommended to carefully check and streamline unnecessary repetitions.
2)The conclusions are a bit wordy and they should be streamlined appropriately
Author Response

(The authors gave the same response as above.)

Reviewer 3 Report
Comments and Suggestions for Authors
This paper conducts a detailed analysis of the causes behind the phase deviation of smartphone antennas in the current state of smartphone positioning and proposes a method for correcting the receiver antenna phase center. There are still some parts that need to be improved and modified. Some main questions are as follows:
1. “The premise of this method is to require the smartphone observation to be able to solve the absolute coordinates of the centimeter level accuracy.”The principles mentioned in part 2.1 of the paper are not mentioned in the following text. What is their role?
2. In Figure 7 and Table 2, what is the reason for the large difference in ambiguity fixing rates under the same environment in adjacent periods?
3. Is the APC coordinates in the smartphone coordinate system universal?
Author Response

(The authors gave the same response as above.)

Round 2
Reviewer 1 Report
Comments and Suggestions for Authors
Authors have responded to this reviewer's in details.
The issues I have raised has been answered in the author's reply. Issues no. 1 to no. 3 have been answered only in the reply, not in the manuscript.
This review thinks that other readers and researchers might have a question regarding the issues 1 to 3 so that it may be better to include some explanations in the manuscript.
Please address the issues 1 to 3 in the manuscript.
Author Response
Thank you again for your comments, and I have uploaded the latest manuscript which adressed the issues 1-3. You can find them in page 3 line108, page 8 line 271, page 14, line 409, accordingly. Please see the attachment.
